# New Concepts for the Diagnosis of Polypoidal Choroidal Vasculopathy

**DOI:** 10.3390/diagnostics13101680

**Published:** 2023-05-09

**Authors:** Jinzhi Zhao, Priya R Chandrasekaran, Kai Xiong Cheong, Mark Wong, Kelvin Teo

**Affiliations:** 1Singapore Eye Research Institute, Singapore National Eye Centre, Singapore 168751, Singaporekaixiong.cheong@gmail.com (K.X.C.); markwongyz@gmail.com (M.W.); 2Tianjin Medical University Eye Hospital, Tianjin 300392, China; 3Ophthalmology & Visual Sciences Academic Clinical Program (Eye ACP), Duke-NUS Medical School, Singapore 169857, Singapore

**Keywords:** non-ICGA, PCV, OCT, color fundus photograph, sub-RPE ring-like lesion, en face OCT-complex RPE elevation, sharp-peaked PED

## Abstract

Polypoidal choroidal vasculopathy (PCV) is a subtype of neovascular age-related macular degeneration (nAMD) that is characterized by a branching neovascular network and polypoidal lesions. It is important to differentiate PCV from typical nAMD as there are differences in treatment response between subtypes. Indocyanine green angiography (ICGA) is the gold standard for diagnosing PCV; however, ICGA is an invasive detection method and impractical for extensive use for regular long-term monitoring. In addition, access to ICGA may be limited in some settings. The purpose of this review is to summarize the utilization of multimodal imaging modalities (color fundus photography, optical coherence tomography (OCT), OCT angiography (OCTA), and fundus autofluorescence (FAF)) in differentiating PCV from typical nAMD and predicting disease activity and prognosis. In particular, OCT shows tremendous potential in diagnosing PCV. Characteristics such as subretinal pigment epithelium (RPE) ring-like lesion, en face OCT-complex RPE elevation, and sharp-peaked pigment epithelial detachment provide high sensitivity and specificity for differentiating PCV from nAMD. With the use of more practical, non-ICGA imaging modalities, the diagnosis of PCV can be more easily made and treatment tailored as necessary for optimal outcomes.

## 1. Introduction

Polypoidal choroidal vasculopathy (PCV), which is characterized by recurrent exudative changes in the macular area and orange–red nodules on clinical examination, was first reported by Yannuzzi in 1990 [1]. PCV is widely recognized as a subset of neovascular age-related macular degeneration (nAMD) and is more prevalent in Asian populations [2,3]. While many aspects of PCV are similar to typical nAMD [4,5], such as abnormal vessel growth, subretinal exudation, or hemorrhage, PCV tends to affect younger individuals compared to typical nAMD [6,7]. The key phenotypic difference between PCV and typical nAMD is the presence of subretinal polypoidal lesions, often with an associated branching vascular network (BVN) [8]. It is essential to differentiate PCV from typical nAMD because there are significant differences in terms of pathogenesis, clinical characteristics, prognosis, and response to treatment [9,10].

Currently, the gold standard for the diagnosis of PCV is indocyanine green angiography (ICGA). This imaging modality can clearly highlight the polypoidal lesion and abnormal BVN from an en face perspective [11,12,13,14]. Despite its ability to detect PCV, there are shortcomings with the use of ICGA. First, the ICGA examination is an invasive procedure with the use of a contrast agent that can result in severe side effects in patients with iodine-containing agent allergies [15]. Secondly, the procedure is time consuming and costly. Therefore, to reduce the reliance on ICGA, much research has been conducted to explore alternative diagnostic methods for PCV. Non-invasive examination methods such as optical coherence tomography (OCT) and OCT angiography (OCTA), color fundus photography (CFP), and fundus autofluorescence (FAF) have been evaluated to assess the accuracy of differentiating PCV from typical nAMD. These modalities may not have the accuracy of ICGA, but are far less invasive, faster, less costly, and can achieve close to gold standard levels of diagnosis for PCV. Non-ICGA diagnosis of PCV is achieved by identifying key structural features that are indicative of PCV on non-ICGA modalities and correlating them with the polypoidal lesion and BVN in ICGA (Table 1).

This review will discuss these methods in detail and evaluate the current evidence for non-ICGA imaging modalities for PCV.

## 2. Optical Coherence Tomography in PCV

OCT measures the light reflection of different tissues and analyzes it with software to form a tomographic image of the internal structure of the tissue [18]. As a non-invasive examination, OCT has been widely used in the diagnosis and treatment of various ophthalmic diseases and has gradually become an indispensable part of the ophthalmic examination [19,20]. OCT technology has advanced from time-domain (TD) to spectral-domain (SD), and swept-source (SS) OCT. Currently, most ophthalmologists use SD-OCT in practice. This technology provides a higher resolution over the older time domain platforms and can detect the structures of the outer retina and choroid with higher resolution. These cross-section images can subsequently be used for qualitative and quantitative analysis of diseases [21,22,23]. The excellent visualization and wide adoption by many retina clinics make SD-OCT a good candidate for diagnosing PCV [24]. Several groups have reported the high diagnostic value for detecting PCV with SD-OCT [16,25,26,27,28].

### 2.1. Classic OCT Features of PCV

Prior studies that established the usefulness of OCT in diagnosing PCV were based on several OCT features that can be correlated with the two main components of PCV that are conventionally seen in ICGA, which are the polypoidal lesion and BVN [4,13,16,17] (Figure 1).

The polypoidal lesion component of PCV that is seen in ICGA has characteristic structural manifestations in OCT. Descriptions of these changes include sharp-peaked PED, thumb-like projections, subretinal pigment epithelium (RPE) ring-like lesion, and notched/multilobular PED. In 2000, Iijima et al. firstly found that the orange–red lesions in the fundus of PCV patients could be seen as sharp protrusions of the RPE in OCT, which were steeper compared with typical serous PED [29]. Osuji et al. believed that the deep and dome-shaped RPE elevations in PCV patients were where the polypoidal lesions were located, with a moderate emission signal or nodular appearance underneath [30]. In 2014, De Salvo et al. named this steep protrusion “sharp peaked PED” and used it as one of the OCT features to differentiate PCV from occult choroidal neovascularization (CNV) in 51 eyes [25]. The sensitivity was 94.6% and specificity was 92.9% [25] (Table 2). In 2016, Liu et al. used the words “thumb-like projection” to describe this PED characteristic and made use of OCT to differentiate PCV from typical nAMD in 156 Chinese patients [16]. They found a sensitivity of 89.4% and a specificity of 85.3% in detecting PCV when at least two of the following three imaging features were present: a thumb-like projection, pigment epithelial detachment (PED), and double-layer sign. 

Some studies have referred to the luminal structures within these PEDs as “hyporeflective lumen” or sub-RPE ring-like lesions and have associated them with polypoidal lesions seen in ICGA [4]. In the Asia-Pacific Ocular Imaging Society (APOIS) PCV Workgroup report, sub-RPE ring-like lesion is listed as one of the three major criteria and achieved an accuracy of 0.91 [28]. Chang et al. used OCT to diagnose 263 eyes of patients with PCV or typical nAMD in a Korean population [26]. When at least three of the following five imaging features were present, including a hyporeflective lumen representing polypoidal lesion, multiple retinal pigment epithelial detachment (RPED), a sharp RPED peak, an RPED notch, and hyperreflective intraretinal hard exudates, PCV could be distinguished from typical nAMD with a sensitivity of 85.7% and specificity of 86.2%. 

In 2007, Tsujikawa et al. discovered that a notched PED was found in 57% of PCV patients in their imaging data analysis using OCT, with most of them showing polypoidal lesions when compared to ICGA [32]. De Salvo et al. also used “notched PED” in 51 patients as one of the OCT features to differentiate PCV and showed a sensitivity of 94.6% and specificity of 92.9% in identifying PCV [25]. In the APOIS report, the panel recommended the term “multilobular PED” to describe the notch within PED and listed it as one of the minor diagnostic criteria [28].

In 2007, Sato et al. reported that 26 out of 44 eyes with PCV exhibited the double-layer sign for the BVN. This is an OCT feature that indicates a shallow and irregular elevation of the RPE from the underlying intact Bruch’s membrane. The upper hyperreflective band of the double layer is of the RPE and the lower band is of the Bruch’s membrane [33]. Sato et al. postulated that the double-layer sign was an accumulation of fluid between the RPE and Bruch’s membrane. However, Ojima et al. believed that the space between the double-layer sign was not fluid, but rather the BVN itself, and speculated that BVN may have characteristics different from neovascularization in typical nAMD [34]. This theory was supported by another study [35], which suggested two types of neovascularization. The source of type 1 BVN is a CNV, and the two high-reflective bands in the double-layer sign are formed by the RPE and Bruch’s membrane [35]. The type 2 BVN is caused by abnormal choroidal vessels, and the two high-reflective bands are formed by the RPE-Bruch’s membrane complex and the inner choroid. This classification is yet to be verified and there is still considerable controversy about the source of BVN [36]. The double-layer sign has been employed in multiple studies as one of the major/minor diagnostic criteria. Liu et al. have also considered the double-layer sign as an independent sign in diagnosing PCV and achieved high sensitivity and specificity, whereas the APOIS has listed double-layer sign as a minor criteria [16,17].

In addition, the association of PCV with focal or diffuse choriocapillaris attenuation and intermediate caliber vessels within the Sattler’s layer that are associated with abnormally dilated Haller’s layer veins has been described. PCV is part of the pachychoroid spectrum of conditions, for which one of the key features is inner choroidal attenuation. Accordingly, APOIS has listed thick choroid with dilated Haller’s layer vessels as one of the minor criteria in the non-ICGA diagnosis of PCV [17]. It is also important to note that a thickened choroid on its own may not have any pathologic consequences and that PCV eyes do not always have thick choroids, as choroidal thickness is influenced by a plethora of systemic and ocular factors [37,38]. Eyes with pachychoroid disease that have normal subfoveal choroidal thickness may instead demonstrate an extrafoveal focality of thickened choroid that colocalizes with the pachyvessels [39].

### 2.2. En Face OCT in PCV

OCT B-scans conventionally provide cross-sectional views of the retina. An en face view is possible and adds a “top down” perspective which can intuitively display the condition of retina surface. The en face OCT can be used to identify the abnormalities in PCV; these include cases of choroidal vessels of a larger caliber with an abnormal configuration, and the polypoidal lesions and the BVN can also be identified occasionally [40]. In particular, the BVN can be detected by a complex RPE elevation on en face OCT. This configuration indicates a neovascular network that connects multiple PEDs, essentially outlining both the BVN and polypoidal lesion. A study conducted in Japan using en face OCT imaging with SS-OCT found that en face OCT confirmed the diagnosis of PCV in eyes that had previously been diagnosed with ICGA. The study correctly identified 84.2% of polypoidal lesions as small round protrusions, and 52.6% of the BVN as smaller elevations of the RPE [41]. Kokame et al. evaluated the diagnostic capability of en face SD-OCT in 100 PCV patients [11]. En face SD-OCT visualized PCV as a subretinal vascular structure with hyperreflective borders and polypoidal dilations, with or without a BVN. It performed as well as ICGA in imaging PCV in 55% of patients; however, it can overestimate the lesion size as it includes areas of PED which may not harbor an active neovascular network [11]. In all, en face OCT allows for the assessment of the pathological structural changes of PCV in the coronal plane and an extensive overview of these changes in a single image. Accordingly, complex RPE elevation on en face OCT has been listed as one of the major criteria for the non-ICGA diagnosis of PCV by APOIS [17].

### 2.3. OCT Guiding Treatment for PCV

OCT is commonly utilized to assess treatment response in PCV by evaluating parameters such as sub/intra retinal fluid (SRF/IRF), intraretinal cysts (IRC), and the presence of PED. These parameters are similar to those used in the assessment of nAMD [42]. In both PCV and typical nAMD, fluid within the retina compartments indicates disease activity and serves as an indication to commence treatment with vascular endothelial growth factor (VEGF) inhibitor therapy. In PCV, the PED characteristics appear to play a more important role in defining disease activity compared to typical nAMD. In typical nAMD, the presence of a PED does not appear to affect visual outcomes [43]. However, in PCV, PED volume specifically was found to be closely associated with visual outcomes. Vyas et al. reported the association of baseline imaging features, including PED parameters (height, volume) and choroidal parameters (choroidal thickness, choroidal volume, and choroidal vascularity index) on visual and anatomical outcomes in eyes with PCV treated with anti-VEGF monotherapy [44]. In the study, PED-related volumetric parameters such as PED volume and PED heights were found to predict disease activity and visual outcomes at month 12 [44].

OCT was also found to be useful in planning for adjunctive rescue photodynamic therapy (PDT) treatment without the need for ICGA in PCV. In PCV, adjunctive use of PDT in combination with VEGF inhibitor therapy appears to provide better outcomes than with VEGF inhibitor therapy alone. Traditionally, ICGA is used to outline the treatment zone for PDT. Teo et al. utilized OCT alone to demarcate the extent of the PCV lesion, both the polypoidal lesions, and BVN [45]. The proposed treatment spot as defined by several retinal experts based solely on OCT was found to cover 100% of the polypoidal lesion area and 91% ± 12% of the BVN area when compared with the conventional spot defined by ICGA [46]. This suggested that PDT could be effectively delivered based on OCT alone with adequate coverage of the entire PCV lesion and its components.

## 3. OCT Angiography in PCV

OCTA is a non-invasive modality to study retinal vasculature. It is based on OCT technology and functions by acquiring volumetric angiographic information by the detection of differences in amplitude, intensity, or phase variance between sequential B-scans from the same area [47,48]. It provides a similar visualization of the retina vasculature to what is provided by traditional dye-based angiography, including ICGA [49].

### 3.1. Comparison of OCTA and ICGA in PCV

Authors de Carlo et al. compared the sensitivity and specificity of ICGA and OCTA (Zeiss Angioplex software) in detecting PCV [12]. PCV was better detected with ICGA in 21 eyes (44.7%) and was better with OCTA in 9 eyes (19.2%) [12]. Among the 44 eyes that had BVN in ICGA, 93.2% also exhibited BVN in OCTA, while among the 28 eyes with polyps in ICGA, 78.6% also showed polyps in OCTA. OCTA signals of the polypoidal lesions were either bright round flow signals or faint round low-flow signals, necessitating the need for ICGA for confirming the location of polypoidal lesions in low-flow cases. However, BVN was readily picked up by OCTA when compared with ICGA (Figure 2). This study highlighted the limitation of OCTA to detect flow below a particular threshold, especially in low-flow polypoidal lesions [12].

Takayama et al. also compared ICGA and OCTA (using RTvue XR Avanti) in detecting polypoidal areas (PA), polypoidal lesions and BVN in PCV, and reported that ICGA showed BVN in 71.4% and polypoidal lesions in 100% of the cases, while OCTA detected polypoidal lesions and BVN in 76.2% and 95.2% of the cases, respectively [50]. This indicated that the detection sensitivity of OCTA for BVN was much more than that of polypoidal lesions. The mean PA was larger in ICGA than in OCTA (*p* = 0.0046) and OCTA could not detect smaller areas of polypoidal lesions. The authors postulated the smaller size detection of PA is mainly due to the different imaging mechanism of ICGA and OCTA. PA was measured in ICGA as dye leakage area outside the choroidal vessel, while OCTA detected signals of erythrocytes motion within the inner diameter of CNV, which was smaller than the outer diameter of the vessels. The authors also suggested that the poor accuracy may be due to the difficulties in detecting blood flow in small or sharply curved vessels in the polypoidal lesions [50].

Seong et al. reported various characteristics of polypoidal lesions using Rtvue XR Avanti OCTA and ICGA [40]. Polypoidal lesions that showed patterns of clusters and aneurysms in ICGA showed a halo (high flow density surrounding the inner regular dark cavity), a vascular network, and rosette patterns (high flow density surrounding the inner irregular dark cavity) in OCTA. The authors also reported that the mean size of polypoidal lesions was greater with OCTA than with ICGA (*p* = 0.014), which contrasts with what was reported by Takayama et al. [31,40,50,51,52,53]. The reason for this discrepancy is unclear. The mean size of BVN was also greater with ICGA than with OCTA (*p* = 0.372) [40].

In another retrospective case series that compared the OCTA and ICGA findings in PCV, PCV was classified into two types using ICGA: polypoidal PCV (type 1 PCV), which is situated in the sub-RPE area, and typical PCV (type 2 PCV), which is located in the choroidal space. Tanaka et al. reported that OCTA detected 17% of type 1 and 46% of type 2 polyps compared with ICGA [53]. The difference in results may be attributed to the segmentation error being just below the RPE. Polyps are often situated beneath the RPE and colocalize with a PED, and PEDs produce a steep slope that makes it difficult to put the sliced line at an appropriate depth. In addition, the presence of a haemorrhagic PED or subretinal haemorrhage increases the difficulty of visualizing a polyp.

### 3.2. Three-Dimensional Anatomical Characterization of PCV Complex by OCTA

A more comprehensive assessment of PCV in OCTA can be achieved by considering the scan in three dimensions. Chi et al. applied a layer-by-layer en face OCTA assessment to study the quantitative three-dimensional characteristics of PCV using the RPE as the reference plane [54]. They observed BVN (55.3%), polypoidal structures (36.2%), and choroidal stalks (origin of the polypoidal complex) (55.3%) in those with PCV. The polypoidal structures are located deeper within the eye than the BVN. BVN was situated 28.6 (14.2) μm below the RPE reference plane, while the polypoidal structures were situated 45.3 (36.1) μm above the RPE reference plane. Stalk-like structures are situated at the outermost position of the PCV complex, separated by a mean of 51.8 ± 16.8 μm from the BVN plane, which may indicate their origin from the choroid vasculature [54].

### 3.3. Patterns of BVN in OCTA

The characteristics of BVN can also be discerned in OCTA. Huang et al. reported several distinct patterns of BVN. The “trunk” pattern, characterized by one or more primary trunks of neovascular vessels with radiating branches extending towards the periphery of the vascular network, was seen in 47% of patients [55]. The “glomeruli” pattern, distinguished by an intensely interconnected vascular network resembling glomeruli in nephrons, but without a discernible major trunk, was seen in 33% of patients. The “stick” pattern, characterized by a small and localized neovascular network without an identifiable feeding vessel, was seen in 20% of patients. The trunk pattern, which consisted of a centrally located major vessel, became more prominent on combination treatment with PDT and anti-VEGF therapy. The glomeruli and stick patterns had higher recurrence rates than the trunk pattern, and the stick pattern was associated with the pachychoroid spectrum disease that was characterized by choroidal hyperpermeability in ICGA [55]. This is consistent with the findings of Pang et al. [56] and Koizumi et al. [57], where the glomeruli and stick patterns had better BCVA and response to combination treatment in comparison with the trunk pattern. They concluded that the difference in response was due to the higher non-foveal involvement in the two other patterns compared with the trunk pattern, although the chances of recurrence were found to be higher. Overall, the authors observed that classifying the BVN could prognosticate visual and anatomical outcomes in PCV.

### 3.4. Accuracy of OCTA in Differentiating PCV from Typical nAMD

Several studies used OCTA to differentiate PCV from non-PCV eyes. Authors de Carlo et al. compared the ability of OCTA to differentiate PCV from typical nAMD and reported a sensitivity of 43.9% and specificity of 87.1% [28]. This level of accuracy was very low and unlikely to be clinically applicable. The reason for this poor differentiation is postulated to be due to the visualization perspective employed by this study. The assessment of OCTA was from an en face view, which is subject to inaccuracies in segmentation, particularly in eyes with PCV where the anatomy can be severely disrupted and automated segmentation algorithms are unable to accurately define the layers of the retina. There could also be significant signal attenuation from a hemorrhage, exudation, PED, and extensive edema, which could further contribute to inaccurate segmentation. In addition, in PCV, there is a large disparity in the location of the BVN and polypoidal lesion, and often, segmentation algorithms are unable to encompass both lesion components, leading to an under-detection of polypoidal lesions [27].

Cheung et al. further refined the use of OCT and OCTA to differentiate PCV from non-PCV in 50 eyes and reported better accuracy; a sensitivity of 82.6%, and specificity was 51.9% [27]. There was a difference in the method used to visualize the polypoidal lesion in OCTA between Cheung and de Carlo’s work. In the work performed by Cheung et al., there was an emphasis on the use of cross-sectional visualization of the flow signal. This meant that the polypoidal lesions, which are often on a different plane from the BVN, could be appreciated better. The authors observed that while a diffuse hyper flow signal in the outer retina was seen in all the eyes with CNV (100%), a localized sub-RPE hyper flow signal was seen in 82.6% of eyes with PCV. This localized sub-RPE hyper flow signal, corresponding to the location of the polypoidal lesion, was also detected in cross-sectional OCT. The localized sub-RPE hyper flow signal was also noted in retinal angiomatous proliferation (RAP) lesions; however, in these lesions, the flow was linear in nature extending from the inner retina to below the RPE [51].

Meanwhile, Inoue et al. used both en face OCTA and cross-sectional OCTA to evaluate the anatomical information of PCV. They observed that en face OCTA provided comparable information about BVN and type 1 CNV but failed to show the morphological characteristics of the polypoidal lesions [51]. In contrast, cross-sectional OCTA not only revealed flow signals within focal regions of the polypoidal lesions, but also showed that other portions of the polypoidal lesion lumen were devoid of flow signal. Several other studies have also corroborated this finding. Srour et al. observed the BVN as a hyper flow lesion on segmentation of the choriocapillaris in OCTA, while the polypoidal lesion was a hyper flow round structure surrounded by a hypo-intense halo or a hypo-flow round structure [52]. Wang et al. showed high flow signals in the outer retina slab (70 µm below the inner plexiform layer to 30 µm below the RPE in the AngioVue software) under the top point of the PED in OCTA in 92.3% of ICGA detected polypoidal lesions [58].

### 3.5. OCTA in PCV after Treatment with anti-VEGF Monotherapy or Combination Therapy

Longitudinal comparison of OCTA imaging post treatment within the same eye is challenging as registration of sequential images and segmentation can vary greatly between time points. Teo et al. in their longitudinal case-controlled study performed serial OCTAs in treatment-naïve PCV with either anti-VEGF monotherapy or combination therapy with anti-VEGF and PDT [45]. In this study, flow signal within the PCV complex in the outer retina, assessment of blood flow in the choriocapillaris, and caliber of the pachyvessel in the Haller’s layer were analyzed, and changes were correlated with that in ICGA and structural OCT. The flow signal of the PCV lesion was significantly reduced in the combination group compared to the monotherapy group (84.6% vs. 40.0%, *p* = 0.04), and flow signal within the choriocapillaris showed patchy reduction in both combination and monotherapy groups (15.4% and 10.0% respectively, *p* = 0.61). Haller’s vessel caliber was significantly reduced in the combination therapy group compared to the monotherapy group (75% vs. 0%, *p* = 0.01). The study showed that OCTA can replace an invasive ICGA during the follow-up of patients after treatment for monitoring and indication of treatment response. The authors postulated that the presence of a persistent BVN and residual flow in the polypoidal lesion might cause recurrence, even though there was no fluid in the retina compartments. In addition, OCTA changes in the Haller’s vessels suggested that the addition of PDT to VEGF inhibitor therapy may not only close the polypoidal lesion, but also modulate the blood flow in the choroidal layer [45].

## 4. Color Fundus Photography in PCV

### 4.1. Classic PCV Features on CFP

Classic features of PCV on CFP include the hallmark orange–red subretinal nodules, spontaneous massive subretinal hemorrhage (SRH, defined as area ≥ 4 disc areas), serous or hemorrhagic PED, serosanguineous maculopathy, absence of drusen, and RPE atrophy [13,59]. However, whilst such features may be highly suggestive of PCV, the presence of such features alone is not diagnostic in the absence of other imaging modalities [59,60]. Instead, CFP is often clinically used in conjunction with other imaging modalities for the diagnosis, differentiation, and monitoring of PCV disease activity [13,61]. This section summarizes the current recommendations that incorporate CFP features for PCV diagnosis and evaluates the utility of CFP (alone, and in conjunction with other imaging modalities) in the diagnosis of PCV.

### 4.2. Role of CFP in Classic PCV Diagnostic Criteria

Classically, there exist two sets of established PCV diagnostic criteria: the Japanese PCV Study Group guidelines, and the EVEREST study criteria [4,62]. Both criteria recommend a combination of CFP and ICGA for PCV lesion detection. The Japanese Study Group defines definitive PCV cases as those with protruded orange–red elevated lesions on fundus examination/CFP and/or characteristic polypoidal lesions in ICGA [61]. Similarly, the EVEREST trial defined PCV based on the presence of focal subretinal hyperfluorescence in ICGA plus at least one of the following criteria: nodular appearance of polypoidal lesions in a stereoscopic examination, hypofluorescent halo around the nodule(s), presence of a BVN, pulsation of polypoidal lesions on dynamic ICGA, orange subretinal nodules on CFP, or a massive submacular hemorrhage (≥4 disc areas in size) [62].

### 4.3. CFP in Non-ICGA Diagnostic Criteria

While CFP holds an essential role in the above two classic PCV diagnostic criteria, recent studies have increasingly focused on non-ICGA PCV diagnostic strategies [17,63]. These studies have evaluated the diagnostic ability of CFP (both alone and in-conjunction with other non-ICGA imaging modalities) to differentiate PCV from nAMD, although they have often concluded that optimal performance comes from a multi-modal approach of combination of CFP features with other imaging modalities, such as fundus fluorescein angiography (FA), OCT, and ICGA.

An initial study by Chaikitmongkol et al. observed that adopting a grader-evaluated CFP-alone approach for PCV diagnosis achieved an area under curve (AUC) of 0.81 but with low sensitivity (0.63) [63]. A subsequent study by the group further assessed specific features of both CFP and OCT for PCV diagnosis [64]. Using a backwards logistic regression model, the authors selected four major diagnostic features; a notched or haemorrhagic PED as detected using CFP (AUC of 0.77), as well as three OCT features: sharply peaked PED at an angle of 70° to 90°, notched or multilobulated PED, and a hyperreflective ring underneath PED. The authors observed that the presence of at least two of these four major features could diagnose PCV with a sensitivity and specificity of more than 90%. Interestingly, the presence of subretinal orange nodules in CFP had a high specificity (0.92) but low sensitivity (0.39) and was therefore not included as part of the major criteria.

Yang et al. adopted a similar approach to evaluate the use of CFP and OCT features to distinguish PCV from typical neovascular age-related macular degeneration (nAMD) within a Chinese population [65]. The authors included five key PCV features in CFP (orange nodule, haemorrhagic PED, multifocal lesions, extensive haemorrhage, and absence of drusen) in their investigation. Of these five CFP features, only the presence of a subretinal orange nodule attained a high accuracy (AUC 0.88). The group eventually derived five major diagnostic criteria: the presence of a subretinal orange nodule in CFP, in conjunction with four OCT features: thumb-like PED, notched PED, bubble sign (hyperreflective bubble-like ring surrounding hyporeflective halo underneath PED), and Bruch’s membrane depression under serosanguinous PED.

The inclusion of subretinal orange nodules in CFP as a major non-ICGA diagnostic criteria by Yang et al. is more in line with the classic diagnostic criterion proposed by the Japanese study group and EVEREST guidelines. Yang et al. reported a sensitivity value of 0.78 for the subretinal nodules, in contrast to the 0.39 in the study by Chaikitmongkol et al. The authors suggested that the differences between the two studies may potentially have been due to different clinical presentations across ethnic groups [66], with the Chinese cohort (Yang et al.) potentially having more common clinical presentations with Japanese patients and the EVEREST cohort in comparison to the Thai (Chaikitmongkol) cohort.

In both ICGA and non-ICGA diagnostic criteria, CFP is important but insufficient. It was interesting that the Japanese study group and Yang et al. both reported high sensitivities for CFP, whilst Chaikitmongkol et al. only reported a much lower sensitivity [61,63,64,65]. We speculate that differences in findings could be due to differences in the study population, or that patients were examined at different stages of the disease process. In the early stages, the orange nodules might have still been masked by blood. The presence of a subretinal red–orange nodule in CFP remains one of the criteria used for both classic and non-ICGA diagnostic criterion, although it should not be interpreted in isolation and is better interpreted when combined with OCT [17,65].

## 5. Fundus Autofluorescence in PCV

FAF is a rapid, non-invasive imaging modality that relies on the inherent fluorescence (autofluorescence) of ocular structures [67,68]. In the production of autofluorescence, fluorophores absorb photons of the excitation wavelength, which elevates an electron to a state of higher energy. The electron then dissipates some energy and emits a quantum of light at a lower energy and longer wavelength as it transitions back to the ground state. These fluorophores include lipofuscin and melanin, and the autofluorescence is visualized with a fundus camera or a confocal scanning laser microscope modified with autofluorescence filters.

There are a few studies which have investigated FAF characteristics of PCV [69,70,71,72,73,74]. The findings are not entirely consistent. Many studies have reported central hypoautofluorescence surrounded by hyperautofluorescence, which indicates RPE atrophy and damage. Correspondingly, prominent anterior protrusions of the RPE which overlie the polypoidal lesions are usually observed in OCT [29,30,75]. The mechanical stress on the RPE layer may result in cellular damage and produce hypoautofluorescence. Histopathologic studies have reported disruptions in the RPE continuity in PCV and attributed this to abnormalities in choroidal vessel haemodynamics [76]. The surrounding hyperautofluorescent ring has been postulated to be a result of the aggregation of FAF from the overlapping RPE at the steeply elevated regions [69]. Another theory for the hyperautofluorescence is that the RPE cells surrounding the central hypoautofluorescence elevate their metabolic rate and accelerate the intracellular accumulation of lipofuscin granules, which manifested as a hyperautofluorescent ring [69].

In a large series of 170 patients with PCV, Zhao et al. reported six autofluorescence patterns, including: 1. confluent hypoautofluorescence with hyperautofluorescent ring (49.8%); 2. confluent hypoautofluorescence (22.6%); 3. hyperautofluorescence with hypoautofluorescent ring (3.7%); 4. granular hypoautofluorescence (7.0%); 5. blocked hypoautofluorescence due to haemorrhage (8.6%); and 6. polypoidal lesions without apparent autofluorescent changes (8.2%). Of 146 BVNs, 97.3% exhibited granular hypoautofluorescence, and others showed blocked hypoautofluorescence due to a hemorrhage [69].

Some of these findings have been reported by other groups as well. Similarly, in another study of 92 patients with PCV, Yamagishi reported two characteristic FAF patterns, confluent hypoautofluorescence at the polypoidal lesions, most of which was surrounded by a hyperautofluorescent ring (80.4% of PCV eyes; 74 eyes), and granular hypoautofluorescence at the BVN (98.9% of PCV eyes, 91 eyes) [70]. In a separate study of 47 patients, Suzuki et al. described confluent hypoautofluorescence surrounded by a hyperautofluorescent ring in 86.1% of eyes, and confluent hypoautofluorescence without a ring in 8.3% of eyes [71]. Öztaş et al., in a study of 29 patients, similarly reported confluent hypoautofluorescence with a hyperautofluorescent ring in 72% of eyes (18 eyes) [72]. The granular hypoautofluorescent FAF pattern was also observed in all 24 BVNs (100%), consistent with the location of the lesions in ICGA [72].

Not all FAF characteristics are reported consistently across studies. For example, Öztaş et al. reported that polypoidal lesions presented with hyperautofluorescence with a hypoautofluorescent ring in 8% of eyes (two eyes), and confluent hypoautofluorescence with a hyperautofluorescent ring in 4% of eyes (one eye) [72]. The former finding was reported by Zhao et al. as well [69], but not in the other studies. The confluent hypoautofluorescence that was reported by Suzuki et al. was without a ring in their study. Zhao et al. also reported two new FAF findings that they found associated with PCV, which were hyper-, hypo-, and hyperautofluorescent circle for PED with surrounding SRF and normal hypoautofluorescent circles for PED alone.

Lai et al., in a study of 32 patients, investigated the FAF properties of pachydrusen in PCV. They reported that the proportion of eyes with FAF abnormality of ≥two disc areas was higher in eyes with PCV than central serous chorioretinopathy (CSCR) (85.3% versus 47.2%, respectively), and that pachydrusen might be an indicator of more severe RPE dysfunction [73].

Regarding longitudinal changes, Yamagishi et al. reported that the elimination of the hyperautofluorescent ring was associated with polypoidal lesion resolution after 12 months (*p* < 0.0001), and further proposed that FAF has a potential to evaluate the therapeutic efficacy of treatments for PCV [74]. Suzuki et al. reported three-year changes in FAF abnormalities. The proportion of eyes with confluent hypoautofluorescence surrounded by a hyperautofluorescent ring decreased from 86.1% to 51.4% (whereas that of eyes with confluent hypoautofluorescence without a ring increased from 8.3% to 43.0%), and that of eyes with no marked changes remained stable at 5.6% [71]. The FAF in 96.2% of resolved polypoidal lesions persisted on images with abnormal FAF during the three years of follow-up. The granular hypoautofluorescence at the BVN at baseline became partially confluent hypoautofluorescence in 85.4% of eyes (41 eyes). The mean area with confluent hypoautofluorescence that corresponded to the BVN increased from 1.75 mm^2^ to 5.10 mm^2^, *p* < 0.001.

Despite having identified the FAF characteristics of PCV, FAF is not typically used in the non-ICGA diagnosis of PCV. Yang et al. investigated the diagnostic accuracy of a few non-invasive multimodal imaging methods, including FAF, in diagnosing PCV and distinguishing PCV from typical nAMD. They reported that additional information from FAF did not improve the predictive accuracy of a diagnostic model that used OCT and CFP features. The FAF features used were hyperautofluorescent ring, and granular hypoautofluorescence [65].

## 6. Fundus Fluorescein Angiography in PCV

Although FA can be used in the diagnosis and classification of typical nAMD, its use in PCV is relatively limited due to its inability to visualize sub-RPE structures, including polypoidal lesions. Chaikitmongkol’s study employed CFP, OCT, and FA as non-ICGA criteria in diagnosing PCV [63]. Although CFP combined with OCT provides high sensitivity and high specificity, adding FA did not improve accuracy [63]. The fluorescein dye has a lower affinity for plasma proteins and leaks more than indocyanine green dye, which can mask the underlying polypoidal lesion and BVN and may cause the size of PCV lesions to be overestimated in FA as compared with in ICGA. Therefore, if FA were to be used to define the treatment zone for PDT in PCV, an unnecessarily larger spot size might be applied. However, FA may be superior to ICGA in marking the lesion area for the assessment of the greatest linear dimension, as ICGA does not permeate PEDs and areas with neurosensory detachments as well [6].

## 7. Multimodal Imaging in PCV

### 7.1. Multimodal Imaging in the Diagnosis of PCV

The evidence summarized above suggests that PCV can be differentiated from typical nAMD without invasive ICGA through the multimodal use of various other imaging modalities.

The APOIS PCV Workgroup set out to provide a non-ICGA diagnostic criteria (Table 3). The study identified 11 non-ICGA characteristics of PCV, and eventually nine were assessed to have a reasonable accuracy: sharp-peaked PED; sub-RPE ring-like lesion; complex or multilobular PED; en face OCT-complex RPE elevation; orange nodule; thick choroid with dilated Haller’s layer; double-layer sign; massive hemorrhage; and fluid compartment. This list was further refined into three major and four minor criteria (Table 3). An AUC of 0.91 was achieved if all three major criteria and any minor criterion were present. If only the three major criteria were used, the AUC achieved was similar at 0.90.

Whilst the initial PCV Workgroup report focused on diagnostic criteria for treatment-naive PCV eyes, the subsequent follow-up report (the APOIS PCV Workgroup report-2) specifically focused on differentiating PCV from typical nAMD in “suboptimal responders” [46]. Such eyes had suboptimal responses to an initial three loading doses of anti-VEGF therapy administered monthly. The rationale behind this work was that most would commence VEGF inhibitor therapy regardless of subtype. Diagnosing PCV might only be necessary in those that did not respond to initial loading treatment. In this report, CFP played a greater role in PCV diagnosis. In contrast to the earlier report which specified three OCT-based major criteria, the report by Teo et al. proposed a different set of major criteria: 1. sharp-peaked PED (AUC = 0.76), 2. hyporeflective sub-RPE ring (AUC = 0.73), and 3. orange nodule on CFP (AUC = 0.67). A combination of these three criteria had a specificity of 0.82 and sensitivity of 0.65 (AUC of 0.85). The presence of orange nodule appeared more useful as a differentiating factor in these suboptimal responders compared with treatment-naive eyes, with Teo et al. suggesting that perhaps such orange nodules (amongst other fundus features) could often be seen more clearly once fluid and blood decrease after initial anti-VEGF treatment.

In 2018, Chaikitmongkol et al. explored the diagnostic efficacy of using different combinations of FA, CFP, and OCT as a multi-modal imaging diagnostic method for PCV [63]. They found that the combination of CFP combined with OCT had the highest efficiency in diagnosing PCV. The presence of at least two of the following four image features suggests PCV: notched or hemorrhagic PED on CFP, notched PED in OCT, sharp-peaked PED in OCT, and bubble sign in OCT. The diagnostic strategy had a sensitivity of 0.95 and specificity of 0.95.

In 2019, Yang et al. explored the possibility of using non-invasive multimodal imaging methods to diagnose PCV [65]. The study found that the diagnosis of PCV could be suggested when at least two of the following five features were present: CFP image with subretinal orange nodules, OCT image with thumb-like protruding PED, notched PED, bubble sign, and Bruch’s membrane depression below hemorrhagic or serous PED. In the validation experiment, the sensitivity of this diagnostic strategy was 0.94 and the specificity was 0.93.

### 7.2. Future Angles—Machine Learning Approaches

In a departure from traditional statistical approaches, recent studies have used a machine learning approach to evaluate the ability of CFP in PCV detection. Rather than focusing on specific, pre-defined, grader-extracted CFP features, these studies used convolutional neural networks to directly process fundus images as the primary inputs [77].

Xu et al. utilized a bi-modal deep convolutional neural network (DCNN) framework to categorize subtypes of AMD and PCV from CFP and OCT images [77]. The DCNN model that was trained on CFP images attained a respectable AUC of 0.929 for PCV detection, although this was lower than the DCNN models trained in OCT (0.931), and a combination DCNN model trained on both CFP and OCT (0.939). The combination DCNN model attained both the highest accuracy (87.4%) and AUC (0.939), and it is the only model exceeding the performance of the best performing retinal expert. Therefore, the authors concluded that while OCT imaging could provide more information about the retina in three-dimensional space, it should not be used alone or take the place of CFP in diagnosing PCV.

It is also worth noting that the deep learning methods using CFP inputs alone managed to attain higher AUCs than the traditional approach based on predefined CFP features [78,79]. This highlights the potential of novel machine learning techniques in elevating the quality of information extracted from CFPs for PCV detection. Both deep learning studies and traditional methods have concluded that optimal performance stems from a combination of CFP and OCT inputs [80].

## 8. Summary

PCV is an important cause of vision loss in elderly and even middle-aged individuals, particularly in an Asian population. Although ICGA is currently considered the gold standard for diagnosing PCV, it is invasive, costly, time consuming, and access can be limited in some settings. Therefore, the use of simple, fast, and easily available non-ICGA imaging methods for diagnosing PCV has significant clinical benefits. The use of OCT appears to be able to diagnose PCV with the greatest accuracy, but the addition of other modalities could serve as confirmatory measures. Accordingly, the use of diagnostic features based on both OCT images and CFP has been shown in multiple studies to be very valuable in distinguishing PCV from typical nAMD. The further development of multimodal imaging diagnostic criteria for PCV is a research goal and direction for future PCV diagnostic standards.

## Figures and Tables

**Figure 1 diagnostics-13-01680-f001:**
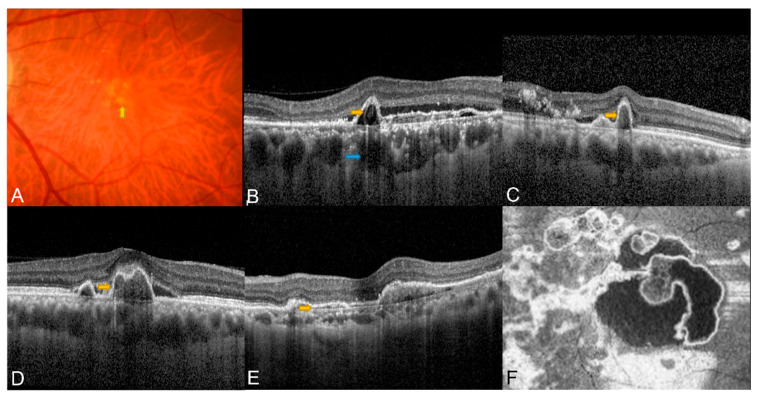
Typical potential diagnostic features detected using CFP and OCT. (**A**) Subretinal orange nodule on CFP (yellow arrow); (**B**) Sub-RPE ring-like lesion (yellow arrow) and thick choroid with dilated Haller’s layer (blue arrow); (**C**) Sharp-peaked/thumb-like PED (yellow arrow); (**D**) Notched/multilobular PED (yellow arrow); (**E**) Double-layer sign (yellow arrow); (**F**). En face OCT complex RPE elevation.

**Figure 2 diagnostics-13-01680-f002:**
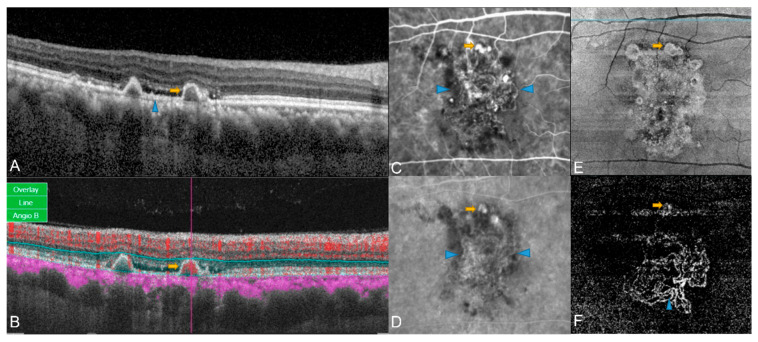
Examples of OCT and OCTA findings in eyes with PCV. (**A**) Structural OCT showing thumb-like PED (yellow arrow) and DLS (blue arrowhead); (**B**) Localized hyperflow sub-RPE signal (red overlay) underneath PED (yellow arrow) observed on cross-sectional OCTA. (**C**,**D**) Early (**C**) and late phase (**D**) of ICGA, both polyp (yellow arrow) and BVN (blue arrowhead) can be seen in ICGA; (**E**) En face OCT showed complex subretinal vascular structure with hyperreflective borders (yellow arrow); (**F**) In the corresponding position on en face OCTA, a nodular hyperflow lesion was seen (yellow arrow), with an adjacent BVN (blue arrowhead).

**Table 1 diagnostics-13-01680-t001:** Terminology of correlated lesion components.

	Year	Author	Terminology	Non-ICGA Modality	Corresponding Lesion Component in ICGA
1	2016	Liu et al. [16]	Thumb-like projection	OCT	Polypoidal lesion
2	2018	Cheung et al. [13]	Orange nodule	CFP	Polypoidal lesion
3	2021	APOIS [17]	Sharp-peaked PED	OCT	Polypoidal lesion
4	2021	APOIS [17]	Sub-RPE ring-like lesion	OCT	Polypoidal lesion
5	2021	APOIS [17]	Multilobular PED	OCT	Polypoidal lesion
6	2018	de Carlo et al. [12]	Notched PED	OCT	Polypoidal lesion
7	2015	Coscas G et al. [4]	Double-layer sign	OCT	BVN
8	2021	APOIS [17]	Complex RPE elevation	En face OCT	Polypoidal lesion+ BVN
9	2018	Cheung et al. [13]	Thick choroid with dilated Haller’s layer	OCT	Pachychoroid

ICGA: Indocyanine green angiography; OCT: Optical Coherence Tomography; CFP: Color Fundus Photography; APOIS: Asia-Pacific Ocular Imaging Society; PED: Pigment Epithelial Detachment; BVN: Branching Vascular Network; RPE: Retinal Pigment Epithelium.

**Table 2 diagnostics-13-01680-t002:** OCT and OCTA-based PCV diagnosis.

	Year	Author	Modality	Features	Standard	Sensitivity (%)	Specificity (%)
1	2014	De Salvo et al. [25]	SD-OCT	Multiple PEDs, sharp PED peak, PED notch, hyporeflective lumen within hyperreflective lesions adherent to RPE	Three or more OCT features	94.6	92.9
2	2016	Liu et al. [16]	SD-OCT	Local PED; DLS; TLP	At least two signs positive	87.5	87.5
3	2016	Chang et al. [26]	SD-OCT	Multiple RPEDs, sharp RPED peak, RPED notch, rounded hyporeflective area representing the polyp lumen within hyperreflective lesions adhered beneath RPE, presence of hyperreflective intraretinal hard exudates	Three or more OCT features	85.7	86.2
4	2017	Cheung et al. [31]	SS-OCTA	Shape; branching; presence of anastomoses and loops; morphology (vascular net with a peripheral arcade vs. a “dead tree” appearance)	All	83.0 (vascular network)	57.1 (vascular network)
						40.5 (polyps)	66.7 (polyps)
5	2019	Cheung et al. [27]	SD-OCT	Notched/narrow-peaked PED; round sub-RPE hyporeflective lesion; RPE detachment (including PED/DLS)	At least 2 signs positive	82.6	51.9
			OCTA	Localized sub-RPE hyperflow lesion in the cross-sectional OCTA; nodular hyperflow lesion in the en face OCTA	At least 1 sign positive	82.6	92.6
			OCT + OCTA	Combined OCT/OCTA diagnosis of PCV	All	82.6	100.0
6	2019	de Carlo et al. [28]	Structural en face OCT	Presence of the PCV complex (BVN and polyps)	All	30.0	85.7
			OCTA			43.9	87.1

Abbreviations: PCV: Polypoidal Choroidal Vasculopathy; SD-OCT: Spectral Domain Optical Coherence Tomography; PED: Pigment Epithelial Detachment; OCT: Optical Coherence Tomography; DLS: Double-layer sign; TLPs: Thumb-like polyps; RPED: Retinal Pigment Epithelial Detachment; SS-OCTA: Swept-source Optical Coherence Tomography Angiography; RPE: Retinal Pigment Epithelium; OCTA: Optical Coherence Tomography Angiography; BVN: Branching vascular network.

**Table 3 diagnostics-13-01680-t003:** Multimodal imaging in diagnosis of PCV.

	Author	Non-ICGA Criteria	Optimal Combination	Sensitivity	Specificity	PPV	NPV	AUC (95% CI)
1	Chaikitmongkol et al. [63]	4 Major criteria 1. Notched or hemorrhagic PED (CFP) 2. Sharply peaked PED (OCT) 3. Hyperreflective ring (OCT) 4. Notched PED (OCT)	≥2 of 4 major criteria	0.95	0.95	0.92	0.95	0.93 (0.89–0.98)
2	Yang et al. [65]	5 Major criteria 1. Subretinal orange nodule (CFP) 2. Thumb-like PED (OCT) 3. Notched PED (OCT) 4. Bubble sign (OCT) 5. Bruch’s membrane depression under serosanguinous PED in OCT	≥2 of 5 major criteria	0.88	0.92	0.79	0.89	0.90 (0.84–0.97)
3	Cheung et al. [17]	3 Major criteria 1. Sub-RPE ring-like lesion (OCT) 2. En face OCT-complex RPE elevation (en face OCT) 3. Sharp-peaked PED (OCT) 4 Minor criteria 1. Orange nodule (CFP) 2. Thick choroid with dilated Haller’s layer (OCT) 3. Complex/Multilobular PED (OCT) 4. Double-layer sign (OCT)	3 Major + any 1 minor	0.78	0.91	0.94	0.68	0.91 (0.86–0.95)

Abbreviations: PCV: Polypoidal Choroidal Vasculopathy; ICGA: Indocyanine green angiography; AUC: Area under curve; CI: Confidence interval; PPV: Positive predictive value; NPV: Negative predictive value; PED: Pigment Epithelial Detachment; CFP: Color Fundus Photography; OCT: Optical Coherence Tomography; RPE: Retinal Pigment Epithelium.

## Data Availability

Not applicable.

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
