# Peer review of "New Concepts for the Diagnosis of Polypoidal Choroidal Vasculopathy"

_diagnostics, 2023, doi:10.3390/diagnostics13101680_

Round 1

Reviewer 1 Report

This is interesting study however I would point out a few major and minor comments.

Major comments:

1.       The association of PCV with a thick choroid and dilated Haller’s vessels with attenuation of the inner choroid has been described, but Authors did not address this issue in the discussion. I would suggest to supplement the OCT chapter with an additional subsection discussing this issue as, according to APOIS, choroidal thickening with dilated Haller's layer was indicated as one of the minor criteria for the diagnosis of PCV.

Minor comments:

Table 2 –1st table row (Da Salvo) – diagnosis based on 3 or more features, while only 3 features were shown in the table. Lack of column headers - applies to all tables.

123. line – the sentence needs rewriting

157.line: Please add a sentence summarizing this subchapter including a critical assessment of the usefulness of en-face OCT. It is also worth noting that the specific features of RPE revealed by this technique have been recognized as one of the main criteria for the diagnosis of PCV according to the APOIS guidelines.

162. line- „fluid within the retina compartments indicate” – indicates (grammatical error)

163. line – „serve as indications” – indication (grammatical error)

191.line:  „Carlo et al. compared ICGA and OCTA (Zeiss Angioplex software) in PCV.” What was compared? Usefulness  both techniques in the PCV diagnosis? Please clarify

192. „In  44.7% of the eyes, ICGA provided better visualization of PCV while OCTA was better in 19.2% of the eyes.” – This statement  does not reflect the results of the cited study. Please rewrite

229. line. Add the definition of type I and II BVN/polypoidal lesion. What was the reason for this discrepancy?

318. line – „VEGG”= VEGF? (typo?)

368. line: ; should be replaced by :

369. line: Please add a definition of "bubble sing"

459. line – „…from AF…” (fluorescein angiography?) or FAF?

520. line – „…of least…” -> of at least

523. and next lines: „This study preliminarily explored the possibility of using multimodal imaging for diagnosing PCV and obtained positive results. However, as mentioned in the study, one of the shortcomings of ICGA as the gold standard for PCV diagnosis is that it is invasive and expensive, with a low rate of equipment accessibility. FA, like ICGA, is also an invasive examination, and its cost is higher than that of non-invasive imaging methods such as CFP, OCTA, and FAF. Therefore, one of the study limitations is that it included FA, which is like ICGA, but did not include other non-invasive imaging methods.” – Please remove this part since it has no importance for the whole context  

547. line:  „…neural networks) to directly…” – no reason for bracket

569. line – „The use of OCT  appears to be able to diagnose PCV with greatest accuracy, but the addition of other modalities could serve as confirmatory measures”. Results of the cited studies indicate that not OCT alone, but OCT in combination with CFP may diagnose PCV with high sensitivity and specificity. I would emphasize the importance of using both techniques together.

The text needs substancial proofreading since it contains grammatical and punctuation errors.

Author Response

Major comments:

Point 1: The association of PCV with a thick choroid and dilated Haller’s vessels with attenuation of the inner choroid has been described, but Authors did not address this issue in the discussion. I would suggest to supplement the OCT chapter with an additional subsection discussing this issue as, according to APOIS, choroidal thickening with dilated Haller's layer was indicated as one of the minor criteria for the diagnosis of PCV.

Response 1: Thank you for your suggestion. We have added an additional paragraph regarding choroidal thickening with dilated Haller's layer as one of the minor criteria for the diagnosis of PCV.

See Line 140 to 150

In addition, the association of PCV with focal or diffuse choriocapillaris attenuation and intermediate caliber vessels within the Sattler’s layer that are associated with abnormally dilated Haller’s layer veins has been described. PCV is part of the pachychoroid spectrum of conditions, for which one of the key features is inner choroidal attenuation. Accordingly, APOIS has listed thick choroid with dilated Haller’s layer vessels as one of the minor criteria in the non-ICGA diagnosis of PCV[17]. It is also important to note that a thickened choroid on its own may not have any pathologic consequences and that PCV eyes do not always have thick choroids, as choroidal thickness is influenced by a plethora of systemic and ocular factors[37, 38]. Eyes with pachychoroid disease that have normal subfoveal choroidal thickness may instead demonstrate an extrafoveal focality of thickened choroid that colocalizes with the pachyvessels[39].

Minor comments:

Point 2: Table 2 –1st table row (Da Salvo) – diagnosis based on 3 or more features, while only 3 features were shown in the table. Lack of column headers - applies to all tables.

Response 2: Thank you for pointing this out. There are four OCT features in the non-ICGA diagnosis of PCV in De Salvo’s paper. We have added the fourth (multiple PEDs) into the manuscript. The author diagnosed PCV vs occult CNV based on the presence of at least three of the following findings: multiple PEDs, sharp PED peak, PED notch, and rounded hyporeflective area representing the polyp lumen within the hyperreflective lesions adherent to the underside of the RPE. The column headers have also been added.

See Table 1. Column headers; Table 2. Row (De Salvo) and column headers; Table 3. Column headers.

Point 3: 123. line – the sentence needs rewriting

Response 3: Thank you for pointing this out. We have rewritten this sentence.

See Line 122 to 124

In 2007, Sato et al. reported that 26 out of 44 eyes with PCV exhibited the double-layer sign for the BVN. This is an OCT feature that indicates a shallow and irregular elevation of the RPE from the underlying intact Bruch's membrane. The upper hyperreflective band of the double layer is of the RPE and the lower band is of the Bruch's membrane [33].

Point 4: 157.line: Please add a sentence summarizing this subchapter including a critical assessment of the usefulness of en-face OCT. It is also worth noting that the specific features of RPE revealed by this technique have been recognized as one of the main criteria for the diagnosis of PCV according to the APOIS guidelines.

Response 4: Thank you for your suggestion. We have added a sentence to summarise the usefulness of en-face OCT.

See Line 168 to 171

In all, en face OCT allows the assessment of the pathological structural changes of PCV in the coronal plane and an extensive overview of these changes in a single image. Accordingly, complex RPE elevation on en face OCT has been listed as one of the major criteria for the non-ICGA diagnosis of PCV by APOIS[17].

Point 5: 162. line- „fluid within the retina compartments indicate” – indicates (grammatical error)

Response 5: Thank you for pointing this out. This is a typographical error. We have corrected this.

See Line 175 to 178

In both PCV and typical nAMD, fluid within the retina compartments indicates disease activity and serves as an indication to commence treatment with vascular endothelial growth factor (VEGF) inhibitor therapy.

Point 6: 163. Line – „serve as indications“ – indication (grammatical error)

Response 6: Thank you for pointing this out. This is a typographical error. We have corrected this in the manuscript.

See Line 175 to 178

In both PCV and typical nAMD, fluid within the retina compartments indicates disease activity and serves as an indication to commence treatment with vascular endothelial growth factor (VEGF) inhibitor therapy.

Point 7: 191.line:  „Carlo et al. compared ICGA and OCTA (Zeiss Angioplex software) in PCV.” What was compared? Usefulness  both techniques in the PCV diagnosis? Please clarify

Response 7: Thank you for your question. de Carlo et al. compared the sensitivity and specificity of ICGA and OCTA (Zeiss Angioplex software) in detecting PCV. We have clarified this in the manuscript.

See Line 205 to 207

de Carlo et al. compared the sensitivity and specificity of ICGA and OCTA (Zeiss Angioplex software) in detecting PCV [12].

Point 8: 192. „In  44.7% of the eyes, ICGA provided better visualization of PCV while OCTA was better in 19.2% of the eyes.” – This statement  does not reflect the results of the cited study. Please rewrite.

Response 8: Thank you for pointing this out. We have changed the citation to “de Carlo, T. E., G. T. Kokame, J. G. Shantha, J. C. Lai, and R. Wee. "Spectral-Domain Optical Coherence Tomography Angi-ography for the Diagnosis and Evaluation of Polypoidal Choroidal Vasculopathy." Ophthalmologica 239, no. 2-3 (2018): 103-09.”

See Line 206 to 207

PCV was better detected with ICGA in 21 eyes (44.7%) and was better with OCTA in 9 eyes (19.2%)[12].

Point 9: 229. line. Add the definition of type I and II BVN/polypoidal lesion. What was the reason for this discrepancy?

Response 9: Thank you for the suggestion and the advice. The reason for the difference in result may be attributed to segmentation error just below the RPE. In addition, the presence of a haemorrhagic PED or subretinal haemorrhage increases the difficulty of visualizing a polyp.

See Line 243 to 251

In another retrospective case series that compared the OCTA and ICGA findings in PCV, PCV was classified into two types using ICGA: polypoidal PCV (type 1 PCV), which is situated in the sub-RPE area, and typical PCV (type 2 PCV), which is located in the choroidal space. Tanaka et al. reported that OCTA detected 17% of type 1 and 46% of type 2 polyps compared with ICGA[53]. The difference in results may be attributed to segmentation error just below the RPE. Polyps are often situated beneath the RPE and colocalize with a PED, and PEDs produce a steep slope that makes it difficult to put the sliced line at an appropriate depth. In addition, the presence of a haemorrhagic PED or subretinal haemorrhage increases the difficulty of visualizing a polyp.

Point 10: 318. line – „VEGG”= VEGF? (typo?)

Response 10: Thank you for pointing this out. This is a typographical error. We have corrected it.

See Line 338 to 340

In addition, OCTA changes in the Haller’s vessels suggested that the addition of PDT to VEGF inhibitor therapy may not only close the polypoidal lesion but also modulate the blood flow in the choroidal layer[45].

Point 11: 368. line: ; should be replaced by :

Response 11: Thank you for pointing this out. This is a typographical error. We have corrected it.

See Line 388 to 392

The group eventually derived five major diagnostic criteria: the presence of subretinal orange nodule on CFP, in conjunction with four OCT features: thumb-like PED, notched PED, bubble sign (hyperreflective bubble-like ring surrounding hyporeflective halo underneath PED), and Bruch’s membrane depression under serosanguinous PED. 

Point 12: 369. Line: Please add a definition of “bubble sing”

Response 12: Sorry that this part was not clear in the original manuscript. We’ve added the definition of bubble sign in the manuscript.

See Line 388 to 392

The group eventually derived five major diagnostic criteria: the presence of subretinal orange nodule on CFP, in conjunction with four OCT features: thumb-like PED, notched PED, bubble sign (hyperreflective bubble-like ring surrounding hyporeflective halo underneath PED), and Bruch’s membrane depression under serosanguinous PED. 

Point 13: 459. Line – „…from AF…“ (fluorescein angiography?) or FAF?

Response 13: Thank you for your question. We meant fundus autofluorescence, which has been abbreviated as “FAF” throughout the manuscript, and not fundus fluorescein angiography (FA).

See Line 479 to 483          

Yang et al. investigated the diagnostic accuracy of a few non-invasive multimodal imaging methods, including FAF, in diagnosing PCV and distinguishing PCV from typical nAMD. They reported that additional information from FAF did not improve the predictive accuracy of a diagnostic model that used OCT and CFP features.

Point 14: 520. Line – „…of least…“ -> of at least 

Response 14: Thank you for pointing this out. This is a typographical error. We have corrected it.

See Line 536 to 538

The presence of at least two of the following four features suggests PCV: notched or hemorrhagic PED on CFP, notched PED in OCT, sharp peaked PED on OCT, and bubble sign on OCT.

Point 15: 523. and next lines: „This study preliminarily explored the possibility of using multimodal imaging for diagnosing PCV and obtained positive results. However, as mentioned in the study, one of the shortcomings of ICGA as the gold standard for PCV diagnosis is that it is invasive and expensive, with a low rate of equipment accessibility. FA, like ICGA, is also an invasive examination, and its cost is higher than that of non-invasive imaging methods such as CFP, OCTA, and FAF. Therefore, one of the study limitations is that it included FA, which is like ICGA, but did not include other non-invasive imaging methods.” – Please remove this part since it has no importance for the whole context  

Response 15: Thank you for your advice. As suggested, we have deleted this section.

Point 16: 547. line:  „…neural networks) to directly…” – no reason for bracket

Response 16: Thank you for pointing this out. We have removed the bracket.

See Line 549 to 551

Rather than focusing on specific, pre-defined, grader extracted CFP features, these studies used convolutional neural networks to directly process fundus images as the primary inputs[77].

Point 17: 569. line – „The use of OCT appears to be able to diagnose PCV with greatest accuracy, but the addition of other modalities could serve as confirmatory measures”. Results of the cited studies indicate that not OCT alone, but OCT in combination with CFP may diagnose PCV with high sensitivity and specificity. I would emphasize the importance of using both techniques together.

Response 17: Thank you for your advice. We have emphasized the importance of using both techniques together in the manuscript.

See Line 575 to 577

Accordingly, the use of diagnostic features based on both OCT images and CFP has been shown in multiple studies to be very valuable in distinguishing PCV from typical nAMD.

Reviewer 2 Report

This study is a well-prepared, didactic / informative and comprehensive review paper. It contains adequate/descriptive tables, pictures and figures from the literature. The purpose /aim  of the study is well analysed and discussed.

The paragraph between 486-490 th lines can be summerized or deleted since it is explained in detail in Table 3 ( Cheung et al  ....)

The chapter of " 7. Multimodal Imaging in PCV", which is between the lines 476 - 543, can be summerized, because some informations and figures  were mentioned in the previous chapters in detail.

No issuses detected regarding english lenguage, and easily can be read and understood

Author Response

Point 1: The paragraph between 486-490th lines can be summerized or deleted since it is explained in detail in Table 3 ( Cheung et al  ....)

Response 1: Thank you for your advice. As suggested by the reviewer, we have summarized this section.

See Line 503 to 510

The APOIS PCV Workgroup set out to provide a non-ICGA diagnostic criteria (Table 3). The study identified 11 non-ICGA characteristics of PCV, and eventually nine were assessed to have reasonable accuracy: sharp peaked PED; sub-RPE ring-like lesion; complex or multilobular PED; en face OCT-complex RPE elevation; orange nodule; thick choroid with dilated Haller's layer; double layer sign; massive hemorrhage; and fluid compartment. This list was further refined into three major and four minor criteria (Table 3). An AUC of 0.91 was achieved if all three major criteria and any minor criterion were present. If only the three major criteria were used, the AUC achieved was similar at 0.90.

Point 2: 2.The chapter of " 7. Multimodal Imaging in PCV", which is between the lines 476 - 543, can be summerized, because some informations and figures  were mentioned in the previous chapters in detail.

Response 2: Thank you for your advice. As suggested by the reviewer, we have summarized this paragraph in a shorten version.

See Line 533

In 2018, Chaikitmongkol et al. explored the diagnostic efficacy of using different combinations of FA, CFP, and OCT as a multi-modal imaging diagnostic method for PCV[63]. They found that the combination of CFP combined with OCT had the highest efficiency in diagnosing PCV. The presence of at least two of the following four image features suggests PCV: notched or hemorrhagic PED on CFP, notched PED in OCT, sharp peaked PED on OCT, and bubble sign on OCT. The diagnostic strategy had a sensitivity of 0.95 and specificity of 0.95.

In 2019, Yang et al. explored the possibility of using non-invasive multimodal imaging methods to diagnose PCV[65]. The study found that the diagnosis of PCV could be suggested when at least two of the following five features were present: CFP image with subretinal orange nodules, OCT image with thumb-like protruding PED, notched PED, bubble sign, and Bruch's membrane depression below hemorrhagic or serous PED. In the validation experiment, the sensitivity of this diagnostic strategy was 0.94 and the specificity was 0.93.